# Machine-Learning Techniques Can Enhance Dairy Cow Estrus Detection Using Location and Acceleration Data

**DOI:** 10.3390/ani10071160

**Published:** 2020-07-08

**Authors:** Jun Wang, Matt Bell, Xiaohang Liu, Gang Liu

**Affiliations:** 1School of Agricultural Equipment Engineering, Henan University of Science and Technology, Luoyang 471003, China; lxh1346057969@163.com; 2School of Biosciences, The University of Nottingham, Sutton Bonington, Loughborough LE12 5RD, UK; matt.bell@nottingham.ac.uk; 3Key Laboratory for Modern Precision Agriculture System Integration Research, Ministry of Education, China Agricultural University, Beijing 100083, China; pac@cau.edu.cn

**Keywords:** dairy cow, estrus detection, location, accelerometer, principal component analysis, machine learning techniques

## Abstract

**Simple Summary:**

We investigated the feasibility of combing location, acceleration, and machine learning technologies to accurately detect dairy cows in estrus. An automatic data acquisition system was developed to continuously monitor the location and acceleration data of cow activities. Estrus indicators were obtained by principal component analysis (PCA) of twelve behavioral metrics generated from the collected data sets, which were: duration of standing, duration of lying, duration of walking, duration of feeding, duration of drinking, switching times between activity and lying, steps, displacement, average velocity, walking times, feeding times, drinking times. We introduced K-nearest neighbor (KNN), back-propagation neural network (BPNN), linear discriminant analysis (LDA), classification and regression tree (CART) algorithms for the estrus identification of cows. A comparative assessment of the integration of algorithms and time windows was performed to for determining the optimal combination. The results achieving in this study suggest that synthesis of location, acceleration, and machine learning methods can be utilized to improve estrus cow detection.

**Abstract:**

The aim of this study was to assess combining location, acceleration and machine learning technologies to detect estrus in dairy cows. Data were obtained from 12 cows, which were monitored continuously for 12 days. A neck mounted device collected 25,684 records for location and acceleration. Four machine-learning approaches were tested (K-nearest neighbor (KNN), back-propagation neural network (BPNN), linear discriminant analysis (LDA), and classification and regression tree (CART)) to automatically identify cows in estrus from estrus indicators determined by principal component analysis (PCA) of twelve behavioral metrics, which were: duration of standing, duration of lying, duration of walking, duration of feeding, duration of drinking, switching times between activity and lying, steps, displacement, average velocity, walking times, feeding times, and drinking times. The study showed that the neck tag had a static and dynamic positioning accuracy of 0.25 ± 0.06 m and 0.45 ± 0.15 m, respectively. In the 0.5-h, 1-h, and 1.5-h time windows, the machine learning approaches ranged from 73.3 to 99.4% for sensitivity, from 50 to 85.7% for specificity, from 77.8 to 95.8% for precision, from 55.6 to 93.7% for negative predictive value (NPV), from 72.7 to 95.4% for accuracy, and from 78.6 to 97.5% for F1 score. We found that the BPNN algorithm with 0.5-h time window was the best predictor of estrus in dairy cows. Based on these results, the integration of location, acceleration, and machine learning methods can improve dairy cow estrus detection.

## 1. Introduction

In mammals, estrus is a behavioral sign that ensures that the female is ready to be mated close to the time of ovulation [1,2]. Standing estrus is often defined as true estrus, when the cow makes no effort to escape when mounted by other cows. Other signs of estrus include mounting of other cows, increased activity, and mucous discharge from the vulva. While standing to be mounted is recognized as the primary behavioral sign of estrus, other behaviors, such as anogenital sniffing, restlessness, bellowing, chin resting, head mounting, and an attempt to mount are considered secondary symptoms [3]. Progesterone measurement in plasma or milk can aid detection of estrus by determining error in other detection methods, such as false positives when using activity [4]. Although the online monitoring device of progesterone concentration is available, it does not fit for these commercial farms highly concerned about profit rates due to the equipment cost and expense of chemicals used per measurement.

The assessment and classification of estrus in dairy cows is directly related to the breeding efficiency, milk production, and economic benefits of dairy farms [5]. A high estrus-detection level can stabilize total pregnancy rate, minimize the interval between calvings, and improve fertilization results. Undetected and falsely detected estrus activity will lead to increased input costs (e.g., artificial insemination, feed) [6]. Therefore, accurate estrus detection is essential to maintain the productivity and reproductive performance of dairy herds.

Commonly, a low-cost approach to estrus detection is implemented on commercial farms by the visual observation of physiological and social behavior patterns [7]. Nevertheless, the development of large dairy herds and all-year-round calving patterns in the dairy-farming industry undoubtedly hinders visual observation [8]. In addition, the extreme decline in estrus duration over the last 50 years, increasing age, milk production, and environmental factors (e.g., higher ambient temperature, uncomfortable housing) can negatively affect length and intensity of estrus expression, leading to the low estrus-detection rate [9]. Hence, a number of automated systems utilizing activity sensors (e.g., pedometers, activity-meters, 3D-accelerometers) have been developed, instead of manual inspection, to monitor the specific changes in a certain kind of estrus-accompanied behavior for promoting the discrimination performance [10]. However, despite the use of estrus detection assisted by activity monitoring technologies, the randomness in the occurrence rate and intensity of a single behavior still makes correct identification difficult [11]. To avoid the false identification of estrus caused by the limited value of using a single behavior, several studies proposed multivariate analysis methods to combine the features of different estrus behaviors. Jónsson et al. investigated a novel detection scheme based on observed distributions of the step count data and the lying balance [12]. Reith et al. stated that the activity and rumination time measured by collar-mounted sensors can significantly increase the sensitivity of estrus recognition [13]. The above-mentioned studies suggest that multi-behavior systems have the potential to improve estrus detection. Accordingly, in the use of multivariate discriminant analysis, the selected combination of behavioral metrics should contain as much effective estrus-related behaviors as possible, and adequately consider the impact of identification algorithms. Additionally, the time-window length is also intimately associated with the detection rate of estrus [14,15]. Consequently, we can improve the estrus detection effect through introducing practical discrimination algorithms based on the combination of multiple behavior parameters and optimizing the time-window length.

The cow’s location in the barn is one of the most direct reflections of the temporal and spatial variation for estrus behaviors. The ultra-wideband (UWB) radio technique has been proven to be accurate in harsh environments, such as those subject to high multipath error and many obstructions, with the possibility for monitoring specific interactions and movements that occur at estrus [16]. Continuous location information achieved by UWB technology can not only advance the accuracy of behavior recognition, but can also provide a basis to analyze the spatial variability of behaviors with centimeter accuracy for estrus detection [17]. The current study tested the hypothesis that activity, location and machine learning technologies can provide higher accuracy for the detection of estrus in dairy cows than is currently available. 

The objective of this study is to assess combining location, acceleration, and machine learning technologies to detect estrus in dairy cows. Four machine learning algorithms (K-nearest neighbor (KNN), back-propagation neural network (BPNN), linear discriminant analysis (LDA), and classification and regression tree (CART)) were used to distinguish estrus cows. Moreover, the impact of algorithms and time-window lengths on detection rate was evaluated by sensitivity, specificity, precision, negative predictive value (NPV), accuracy, and F1 score.

## 2. Data Acquisition System

Our system contained a neck tag, eight anchor nodes, a receiver, and a laptop. The neck tag enabled the automated measurement of both location and acceleration of cow behaviors every second. The neck tag was mainly composed of a low-power microprocessor (STM32F103C8T6, STMicroelectronics Ltd., Geneva, Switzerland), a wireless transceiver (DW1000, Decawave Ltd., Dublin, Ireland) based on UWB, and a high-resolution accelerometer (ADXL345, Analog Devices Inc., Norwood, MA, USA). The neck tag was placed in a water-resistant plastic bag. The power supply consisted of two 3.7 V lithium ion batteries (ARB-L18-3500, FENIX Ltd., Shenzhen, China). The protected tag was inserted into a plastic case (102 × 63 × 53 mm) equipped with adjustable straps, and the total weight was approximately 200 g. The adjustable straps facilitated a proper fit of the neck tag to the dimension of cow’s neck in order to have the *y*-axis of the coordinate system of the neck tag aligned with the upward direction perpendicular to the ground (Figure 1). Furthermore, the 0.25 kg iron counterweight was worn below the neck for maintaining neck tag stable. The anchor node was identical to the neck tag, except that it was not equipped with an accelerometer and batteries. Each anchor node was powered by an AC/DC adapter and attached to a pole at a height of 1.8 m using cable ties. The position of Anchor 1 was defined as the origin of the plane coordinate system in the barn. 

The distances between neck tag and anchor nodes installed at the boundary of studied area (Figure 2) were calculated by double side two way ranging (DS-TWR) method [18], and the two-dimensional location data of the neck tag can be solved by adopting trilateration localization algorithm. Acceleration data were synchronously recorded in three dimensions (*x*-, *y*- and *z*- axis) of cow movements (range from −2 to +2 gravity). Both location and acceleration data were wirelessly transmitted with a frequency of 1 Hz to a receiver placed at a distance of about 10 m outside the monitoring area, and connected to a laptop with a USB to TTL module (PL-2303HX, Prolific Technology Inc., Taiwan). A laptop was used to gather incoming data, which were processed, analyzed to generate an estrus alert for individual cows using software on the laptop, or sent to a mobile device. 

## 3. Experimental Setup and Data Collection

During our research, all animals were kept in a pathogen-free environment and fed naturally. The procedures for care and use of animals were approved by the Ethics Committee of the Henan University of Science and Technology, Luoyang, China. All of the experimental procedures were conducted in conformity with institutional guidelines for the care and use of laboratory animals at Henan University of Science and Technology and with the National Institutes of Health Guide for Care and Use of Laboratory Animals (NIH Pub. No. 85-23, revised 1996).

The experiment was undertaken in August 2019 for 12 days at the dairy unit of Sansege Dairy Co., Ltd. (33°05′50.64″ N, 112°32′25.32″ E), Nanyang, Henan Province, China. The study pens had a rectangular layout of 66 × 14 m in an east-west direction and included a drinking area, a row of self-locking headlock, and a row of head to head stall arranged with sand beds. The roof adopted symmetrical structure covered with steel trusses and fiber-board supported by purlins. The height of the barn and the eaves were 10 m and 4.65 m, respectively. Cows were milked twice a day at 5 a.m. and 5 p.m., and concrete floors were cleaned every 4 h with scraper plates. With the assistance of an experienced veterinarian, an unpregnant cow was randomly selected from 353 open lactating Holstein cows in the herd at 7 a.m. each day, and placed in the studied area for 6 h of monitoring without external interference from 8:00–11:00 a.m., and also 1:30–4:30 p.m. (Figure 3). 

A total of 12 cows were chosen during the study period (Table 1). The statistical power of the experimental samples with a significance level of 0.05 was 0.676, which indicated there was a 67.6% or greater probability that the differences between samples can be correctly identified. Meanwhile, whether the cow was in estrus is determined by the standard criterion detailed in Section 3.1. Due to data packet loss and accidental network delays, a total of 25,684 sets of original data were obtained, of which 23,456 sets of data with the duration of movement behavior exceeding 2 s were used for the analysis of estrus recognition algorithms.

### 3.1. Estrus Observations

To determine which cows are in estrus, the cows were observed to detect spontaneous behavioral estrus by applying four video cameras (SNC-VB640, Sony Corporation, Tokyo, Japan) for approximately 18.32 ± 1.5 h/d. These cameras were fixed at pillars about 4.5 m above the trial area in which the cows were housed, to prevent disturbing the animals. The cameras were connected to an external hard-drive video recorder (K-NL408K/CH, Panasonic Corporation, Osaka, Japan). Among all of the possible views available from the video system, plan views were the most appropriate here. A top view of the system provided a panoramic rectified image of the area of interest at a resolution of 1920 × 1080 pixels. Cows were plainly verified by an individual combination of colored tape on each cow. Two experienced observers differentiated estrus cows according to Van Eerdenburg et al. (1996) through retrospectively reviewing video recordings and recording their behaviors [19]. The Intra-class correlation coefficient (ICC) of inter observer reliability was 0.997. Each behavior associated with estrus was allocated a certain number of points (Table 2). If the sum of points during observation period exceeded 100, the animal was regarded to be in estrus.

According to the time-window length of 0.5 h, 1 h, and 1.5 h, we divided the acquired valid data into 144 groups, 72 groups, and 48 groups in accordance with the sampling sequence. Each group of data for each dairy cow corresponded to a judgment result as estrus status label by the standard criterion of estrus (Table 2). Afterwards, we utilized the cow behavior analysis method for location and acceleration data proposed by Wang et al. (2018) to process the grouped data [20]. Through the classification of behavior patterns, each group of data was transformed into twelve behavioral metrics associated with estrus (Kerbrat et al. (2004) and Aungier et al. (2015)): duration of standing, duration of lying, duration of walking, duration of feeding, duration of drinking, switching times between activity and lying, steps, displacement, average velocity, walking times, feeding times, drinking times [21,22]. The acquisition processes of behavioral metrics were classified into two categories. The steps, displacement, and average velocity were calculated by the continuous positioning data of monitored cows in every time window. For the other behavioral metrics, the acceleration data were firstly used to distinguish the behavior patterns of standing, lying, walking, feeding, drinking, lying down, and standing up by the BP-AdaBoost algorithm. Furthermore, applying the D-S evidence theory to fuse the belief assignment functions of behavior categories and location data, to improve the accuracy of behavior discrimination, and then the behavioral metrics were acquired by counting the duration or number of behaviors in an individual time window.

To withdraw dimension problems and difficulties in using data sets originated by over-comprehensive variables and overlapping description information, we adapted SPSS statistical program (SPSS 25.0 for Windows) to conduct principal component analysis (PCA) on all the reconstructed data groups for different time-window lengths. The Kaiser-Meyer-Olkin statistics (KMO > 0.7) and Bartlett’s Test of Sphericity (BTS < 0.05) were applied to carry out PCA adaptability inspection on the data [23]. Moreover, in terms of the principle that the characteristic root was greater than 1, and the cumulative contribution rate was higher than 85%, appropriate principal components were selected as the estrus indicators. We then split the obtained data groups of each time-window length, after PCA dimensionality reduction processing, into training and testing sets for estrus recognition algorithms. Seventy percent of the data was selected as the training set, and the remaining 30% was used for the testing set. 

### 3.2. Estrus Recognition Algorithms

In this study, four machine learning algorithms from SPSS, KNN, BPNN, LDA, and CART, were used to identify the estrus of dairy cows to select the optimal recognition method.

#### 3.2.1. KNN

The KNN method is an instance-based learning method that stores all available data points and classifies new data points based on similarity measures [24]. The idea underlying the KNN method is to assign new unclassified examples to the class to which the majority of its *K* nearest neighbors belongs. As a result of the advantage of decreasing the misclassification error, the KNN algorithm has been used in applications such as data mining, statistical pattern recognition, image processing, and so on. We used the KNN method to classify the estrus states of cows, set Euclidean distance as the distance measure function, and realize the autonomous optimization of *k* value within the range of 1–9, according to the 10-fold cross-validation of the training set and testing set.

#### 3.2.2. BPNN

The BPNN model is made up of various layers of nodes, and is designated by the node characteristics, network interconnection geometry, and the learning rules (transfer functions). Learning is fed back into the model continuously to modify the weights of the nodes between layers to diminish the errors between the predicted and measured data [25]. After determining the weights of the node through the training process, the BPNN model can be practiced for pattern identification with new data. We utilized the BPNN model to distinguish the cows in estrus, adopting the estrus indicators as the input layer and the estrus status label as the output layer. In the meantime, a hidden layer was built by the use of the automatic architecture in SPSS, and the adjustment mode of the number of hidden layer nodes was set to the best estimation accuracy. The activation functions of the hidden layer and the output layer were set as hyperbolic tangent and identity, and a cross-entropy function was selected as the error function. The maximum iteration times, training objectives, and learning efficiency were chosen as 1000, 0.00001, and 0.01, respectively. Additionally, the optimization algorithm applied the conjugate gradient method.

#### 3.2.3. LDA

The LDA algorithm is a well-known method for dimensionality reduction and classification that projects high-dimensional data onto a low-dimensional space, where the data accomplishes maximum class separability [26]. The derived features in LDA are linear combinations of the original features, in which the coefficients are from the transformation matrix. The optimal projection or transformation in classical LDA is obtained by minimizing the within-class distance and maximizing the between-class distance simultaneously, thus, to achieve the maximum class discrimination. It has been implemented successfully in many applications, including pattern recognition and data analysis. For detecting estrus cows by the utility of LDA, we selected Fisher’s discrimination criterion in multivariate analysis, and arranged the prior probability according to the sample size of estrus and non-estrus cows in the training set. Moreover, the Wilks Lambda statistic and leave-one-out cross-validation were employed, to assess discriminant value and discernment performance.

#### 3.2.4. CART

With the use of tree-building algorithm, CART operates by recursively splitting the data until ending points, or terminal nodes, are obtained using preset rules [27]. The CART technique essentially consists of an analytical process that the relative significance of each factor is evaluated and an integral process involving the identification of the optimal combination of independent variables over the dependent variable is utilized. CART has been widely conducted with a high level of accuracy and performance for classifying and forecasting problems. We established a CART model to monitor the estrus situations of cows through SPSS, set the maximum tree depth as 5, the minimum cases in parent node as 7, and the number of child nodes as 3. By applying the Gini index as an attribute selection measure, the nodes were split, and the decision tree was pruned. Meanwhile, the maximum difference in risk generated during pruning was set as 0.

## 4. Data Analysis

### 4.1. Positioning Performance of the Neck Tag

The positioning performance of neck tag was associated with the statistical accuracy of behavioral metrics and directly determined the detection effect of the estrus recognition algorithms. We implemented a series of trials to analyze the exactness and usability of location information. 

A lattice of 25 × 25 cm squares was laid on the passages and alleys of the studied area. Under the guidance of an experienced veterinarian, a cow equipped with the neck tag was led to walk freely in a counterclockwise direction for three times along these passageways. The movement route of the cow was recorded by combining with manual observation and the video system. The real position of a cow was determined as the x and y coordinates of the center of square, in which the vertical projection of its neck tag on the ground was located. The measured location of a cow was the 2D-position information collected by the data acquisition system designed in this study. We used the positioning data of six static points with the residence time of more than 3 min and ten dynamic points during the whole walking period to compare with the real locations for appraising the positioning performance of the neck tag. We calculated planimetric location error as:(1)$$\varepsilon_{i}=\sqrt{{\left(x_{i}-x_{i}^{T}\right)}^{2}+{\left(y_{i}-y_{i}^{T}\right)}^{2}}$$
where *ε_i_* was planimetric location error, (*x_i_*, *y_i_*) was the real position of the measured point, and $\left(x_{i}^{T},y_{i}^{T}\right)$ was the estimated position of the measured point gained by the devised system. Identification and filtering of anomalous location measurements, which were highly different from the central data distribution values, were carried out by adopting an outlier data cleaning technique [28]. The measurements higher than $q_{3}+w(q_{3}-q_{1})$ and lower than $q_{1}-w(q_{3}-q_{1})$, where *q*_1_ and *q*_3_ were the 25th and the 75th percentiles, respectively, and *w* = 1.5, were discarded. After the completion of data cleaning, we judged the positioning performance of the designed system through utilizing the minimum error, maximum error, mean location error, standard deviation, and coefficient of variation for these measurement points.

To evaluate the positioning capability of the data acquisition system, the accuracy and error rate were calculated as follows:(2)$$\mathrm{error}\;\mathrm{rate}=\frac{\mathrm{number}\;\mathrm{of}\;\mathrm{false}\;\mathrm{positives}}{\mathrm{number}\;\mathrm{of}\;\mathrm{true}\;\mathrm{positives}+\mathrm{number}\;\mathrm{of}\;\mathrm{false}\;\mathrm{positives}}$$
(3)$$\mathrm{precision}=\frac{\mathrm{number}\;\mathrm{of}\;\mathrm{true}\;\mathrm{positives}}{\mathrm{number}\;\mathrm{of}\;\mathrm{true}\;\mathrm{positives}+\mathrm{number}\;\mathrm{of}\;\mathrm{false}\;\mathrm{positives}}$$
where the number of true positives was achieved by counting the number of times that the cow was correctly located by the system, the number of false positives was obtained by computing the number of times that the cow was wrongly positioned by the system. The planimetric location error within the acceptable range (mean location error − 1.96 × standard deviation, mean location error + 1.96 × standard deviation) was considered as a correct positioning, and vice versa. In addition, two performance metrics (Metric A, and Metric B) were built for ensuring data consistency before and after cleaning.

*Metric A*: for the monitored points, all the *ε_i_* were used to compute the mean location errors and the related standard deviations. In this metric, the true positives were assigned to these *ε_i_*. Therefore, the number of true positives was obtained by counting the number of *ε_i_*. The number of false positives was assumed to be equal to 0. 

*Metric B*: for the monitored points, all the *ε_i_* obtained after the outlier data cleaning process were applied to calculate the mean location errors and the related standard deviations. In this metric, the number of true positives was obtained by summing the number of *ε_i_* that were not filtered out by the data cleaning process. The number of false positives was presumed to be equal to the number of measurements considered as outliers. 

### 4.2. Estrus Detection

The detection if estrus was classified as either positive (the modeled estrus status) or negative (non-estrus status). We labeled misclassifications of negative and positive samples as false positives and false negatives, respectively. We evaluated the performance of these four machine learning algorithms based on sensitivity, specificity, precision, NPV, accuracy, and F1 score (Table 3). 

## 5. Results

### 5.1. Localization Effect of the Neck Tag

Table 4 shows the minimum errors, maximum errors, mean location errors, standard deviations, and coefficients of variation for planimetric location errors of neck tag at static and dynamic points obtained before and after outlier data cleaning. Before the application of outlier data cleaning, the average of maximum errors and mean location errors of neck tag at six static points were 0.76 m and 0.2617 m, respectively. By eliminating the unreliable data which accounted for about 4.3% of the sets of planimetric location errors, the average of maximum errors and mean location errors changed to 0.425 m and 0.2467 m, respectively. Furthermore, the most significant reduction of maximum errors and coefficients of variation were 0.63 m and 0.26 m, separately. The data processing on dynamic points had the same impact. It is distinctly observed that this method can dramatically diminish the abnormal positioning caused by signal occlusion and the offset of neck tag position.

Figure 4 exhibits the boxplot of planimetric location errors of neck tag at static and dynamic points. It can be noticed that the outlier error range at static and dynamic points was from 0.44 m to 1.04 m, and from 1.02 m to 1.48 m, respectively. The range of the lower quartile of neck tag at static points was from 0.158 m to 0.233 m, and the average value was 0.199 m. The value of the upper quartile was between 0.247 m and 0.353 m, with an average value of 0.299 m. Moreover, it was remarkable that the upper and lower quartiles of neck tag at dynamic points were 2.04 times and 1.75 times of the average value at static points, respectively.

Table 5 depicts the positioning performance of neck tag before and after outlier data cleaning. The acceptable errors after outlier data cleaning (AEAODC) of neck tag data at static and dynamic points were lower, compared with the positioning errors of before outlier data cleaning (BODC), and after outlier data cleaning (AODC). In three stages, the error rate and precision of BODC were the best, and AODC had a good tradeoff between the evaluation criteria and statistical parameters of the location effect. The difference between BODC and AODC in error rate and precision principally stems from the assumption that the number of error positions of the former is 0 according to Metric A, and that the latter regards the outlier data as the false position data by Metric B. Moreover, the high similarity between AODC and AEAODC in positioning performance is chiefly due to the normal distribution characteristics of the planimetric location error data, and the impact of acceptable error range of AEAODC cannot generate a notable difference.

### 5.2. PCA Preprocessing of Behavioral Metrics

Figure 5 displays the change of characteristic roots and cumulative contributions of principal components in three-time windows. As can be seen from Figure 5, the influence of time-window lengths on the characteristic root and cumulative contribution rate of each principal component was not significant. Through the proposed method of principal component selection, the first four principal components (PCA1, PCA2, PCA3, PCA4) were determined for every time-window length. We successfully reduced the dimensions of twelve behavioral metrics and established the estrus indicators.

Table 6 presents the loading scores of the first four principal components in twelve behavioral metrics for the time-window length of 0.5 h, 1 h, and 1.5 h. It can be observed that the correlation between principal components and behavioral metrics was similar for the time-window length of 0.5 h and 1 h. PCA1 had approximate positive loads with larger values for duration of standing, duration of lying, duration of walking, steps, switching times between activity and lying, displacement, and average velocity, which verified the increased activity level of a dairy cow in estrus. In PCA2, the positive loads in duration of feeding and feeding times were dominant, the loads in duration of drinking and drinking times for PCA3 were considerably higher than other factors, and PCA4 had no distinct load tendency. While at the 1.5-h time window, the correlation exchange between PCA2 and PCA3, and PCA1 and PCA4 remained unchanged.

### 5.3. Performance Evaluation of Algorithms for Estrus Detection

Figure 6 manifests the performance of the four machine-learning algorithms for estrus detection. It can be seen that the sensitivity of KNN and BPNN for different time-window lengths was more than 88%, and the sensitivity of LDA and CART fluctuated from 75.81% to 78.95%, and from 73.33% to 86.67%, respectively. In addition, KNN, BPNN, and CART all showed the maximum sensitivity at the 0.5-h time window. Compared with other algorithms, the sensitivity of BPNN was the highest for each time window length. For the 1-h and 1.5-h time windows, the specificity of KNN, BPNN, LDA, and CART were 66.67% and 66.67%, 85.71% and 82.5%, 73.68% and 63.64%, and 71.43% and 80%, respectively. BPNN was the largest in terms of specificity. At the 0.5-h time window, the specificity of KNN (50%), BPNN (53.33%), and CART (54.55%) was notably lower than that of LDA (74.36%). It was discovered that the specificity of LDA decreased with the increase of the time window length, and the other three methods improved with the rise in the time window length. Moreover, with the change of time window, the precision fluctuation range of BPNN, LDA, and CART was 2.5%, 3.51%, and 5.02%, respectively, while that of KNN was the largest, which indicated that BPNN had the most dependable performance in terms of precision. 

## 6. Discussion

As one of the main reasons for the degradation of cow reproductive performance, the importance of estrus detection is self-evident. A large number of studies have revealed that abnormal behavior and time allocations of cows in estrus can be employed as an indicator of estrus detection. On this basis, various estrus automated detection systems have been developed. Rorie et al. used a rump-attached detector and obtained an accuracy of 87.5% by monitoring mounting activity [29]. Brehme et al. applied a combination of sensors to determine activity, lying time, and temperature, and detected estrus with 90% accuracy [30]. The current study also validated the indicative effect of using behavioral changes to detect estrus. According to the estrus indicators established by twelve behavioral metrics, the average sensitivity, average specificity, average precision, average NPV, average accuracy, and average F1 score identified by the four machine learning algorithms were 85.27%, 68.54%, 86.75%, 70.38%, 81.12%, and 85.83%, respectively. The average sensitivity, average precision, and average F1 score (all above 85%) performed well, which was because the selected combination of behavior categories can contain more extra estrus information and provide a more accurate discrimination basis. Nevertheless, the average specificity and average NPV (both less than 80%) were much lower. The two parameters should be further improved in terms of false-positive alerts and false-negative alerts caused by unclear differentiation of estrus or non-estrus behaviors, which can be produced through time-window length, the original information missed by PCA, and estrus intensity. In addition, the specificity and NPV of BPNN in the 1-h time window were 85.17% and 80.75%, respectively. It can be noticed that the difference in algorithm selection and monitoring interval is additionally an essential factor affecting the performance of these two parameters. Moreover, accuracy is influenced by sensitivity, specificity, precision, and NPV. The average accuracy of the four algorithms reached 81.12%, and the detection accuracy of BPNN with the best performance was more than 88% under different time windows.

Cow location information has a definite value for individual identification, behavior classification, and feeding management. Numerous studies have been conducted on cow positioning. Porto et al. proved that the real-time location system with UWB technology could acquire an average position error of 0.11 m in the semi-open barn [31]. By utilizing the indoor positioning system based on ultra-high frequency (UHF) technique, Ipema et al. achieved a static position error of 0.3 ± 0.25 m in the barn [32]. In this study, UWB technology was used to obtain the planimetric coordinates of the cow, and the eight anchor nodes were deployed based on the rectangular partition to accomplish the reliable coverage of radio frequency signals in the studied area. At the same time, we also performed some improvements in the positioning mechanism. Firstly, the neck tag was set to communicate with the nearest four adjacent anchor nodes in turn for avoiding the possible ranging failure affected by channel congestion. Secondly, the ranging correction based on signal power and antenna delay correction were adopted to diminish the range measurement error caused by the obstruction of the housing facilities. Additionally, compared with the radio frequency identification (RFID) system integrated with UHF, UWB technology can enhance positioning accuracy through short pulse signal propagation. The final results showed that the static error and dynamic error of the positioning method practiced in this work were 0.25 ± 0.06 m and 0.45 ± 0.15 m. The smaller positioning error can adequately record the behaviors of each cow in real time.

Multiple studies on cow estrus detection using behavioral metrics mostly used statistical analysis methods. Moore et al. identified 55% of visually observed estrus by comparing the variation of mean daily activity [33]. Jónsson et al. achieved a sensitivity of 88.9% using the means from statistical change detection [34]. Machine learning methods can describe multiple complex interaction relationships or nonlinear relationships, and, thus, bring about remarkable predictive accuracy. Therefore, machine learning methods are suitable alternatives to statistical approaches for automated detection systems of cow estrus, as they focus on the prediction and ability to discriminate behavioral signs. In this study, each set of estrus indicators extracted from the acquisition data sets grouped by monitoring intervals was utilized as the input of a training set, and the estrus status of the corresponding cow determined by the visual observation was adopted as the output of the training set. Afterwards, through the learning of constructed training sets, KNN, BPNN, LDA, and CART algorithms were modeled to identify cows in estrus, and the accuracy of these machine learning methods would be improved with the increase in the number of training sets. Moreover, the prospect of machine learning technology is to perceive the characteristics of estrus activities of dairy herds for enhancing the accuracy in large scale application, and the training process does not rely on the estrus data of an individual cow. This study verified the estrus detection performance of KNN, BPNN, LDA, and CART algorithms, with the accuracy range of 72.73% to 95.36%, and the average sensitivity of 90.83%, 93.12%, 77.14%, and 80%, respectively. We preliminarily presumed that BPNN had the most reliable estrus detection effect, followed by KNN, CART, and LDA. Furthermore, the performance of the four algorithms in terms of accuracy and F1 score proved our judgment on the ranking of the estrus detection capability of these algorithms. For NPV, BPNN was incomparably ahead of the four kinds of algorithms in each time length. Therefore, we determined that the BPNN algorithm with the 0.5-h time window was the optimal detection method.

The difference of implementation principle may chiefly offer the disparity in estrus detection rate of these algorithms. For example, BPNN and CART achieved the accuracy of 95.36% and 72.73% in the time window of 0.5 h and 1 h, respectively, which was because of the result that the BPNN algorithm obtained the optimal weight of each layer through the forward propagation of signal and the backpropagation of error. Although CART is easy to understand and analyze visually, the risk of underfitting of the pruned decision tree is one of the possible reasons for the low accuracy due to the measures of pruning and limiting the depth of the decision tree, to prevent overfitting. In general, compared with statistical methods, these four machine learning methods have a more robust ability to mine and interpret estrus behaviors, and can dependably predict emerging data under the estrus discrimination rules summarized in the learning phase. However, the online identification and detection accuracy of these four algorithms still need to be confirmed by a large number of animals in the future.

Various behaviors have different levels of indicating function on cow estrus. Accordingly, in the studies of estrus detection based on behavioral signs [35], the choice of behavioral metrics is crucial to enhance the detection level. For instance, Valenza et al. recognized 66% estrus events by the pressure-activated Heatmount detectors affixed midline to the rump between the tail head and the tuber coxae for all cows in the experiment [36]. In this study, twelve behavioral metrics were reduced to estrus indicators by PCA, which not only guaranteed the comprehensiveness of temporal and spatial information of estrus behaviors, but also effectively restricted the data scale of algorithm training. Compared with the estrus detection methods using single behavior or several behaviors, this study had more advantages in reducing the misjudgments generated by the short-term abnormal behavioral changes during estrus, under the premise of considering the influence of the time window.

The interval for monitoring is intimately associated with the detection rate of estrus, which is critical to monitoring short-term abnormal behavioral variations and the stability of long-term detection. By utilizing sensitivity, specificity, precision, NPV, accuracy, and F1 score, this study assessed the detection performance of KNN, BPNN, LDA, and CART algorithms in 0.5-h, 1-h, and 1.5-h time windows, respectively. Through the two-factor analysis of variance, it can be discovered that the time window and its combination with algorithms had significant effects on specificity and precision. Meanwhile, the algorithm type also had a considerable impact on precision. In addition, the time window had no vital influence on other parameters (*p* > 0.07), but these parameters were significantly affected by algorithm types. The impact of these factors on enhancing the detection rate should be regarded while establishing the estrus detection model. 

Although we evaluated the estrus detection capability of the developed neck tag for 12 Holstein cows, further trials should be carried out by increasing the number of enrolled cows and deployment scale. The optimal combination of algorithm and time window proposed in this work still needs to take into account the influence of factors such as milk yield and parity on the detection applicability. The next step in the development of this system is to advance real-time analysis functions. Once the functionality and reliability have been confirmed on a larger scale, commercialization is possible.

## 7. Conclusions

Measurements of location and acceleration information obtained with the neck tag proved to be acceptable for the conditions of this study when cows were housed in the barn. However, data packet dropout and unexpected network delay were observed. The results of the proposed estrus detector based on machine learning techniques showed improved performance, an enhanced number of successful alerts, and a reduced number of false positives compared to statistical analysis methods. The BPNN algorithm with a 0.5-h time window achieved the ideal identification performance. Our results suggest that the combination of the data acquisition system and machine learning methods is an alternative to visual observations for indoor-housed cows. Furthermore, the use of the PCA in dimension reduction of correlated behavioral metrics should be advised for the determination of estrus indicators. The estrus indicators originated from the location and acceleration data and the appropriate time window were verified for the positive effects on detection rate of estrus. 

## Figures and Tables

**Figure 1 animals-10-01160-f001:**
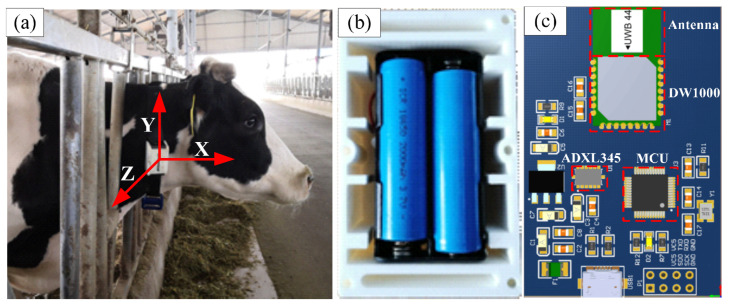
The neck tag for data acquisition (**a**) Coordinate system (**b**) Batteries (**c**) PCB layout.

**Figure 2 animals-10-01160-f002:**
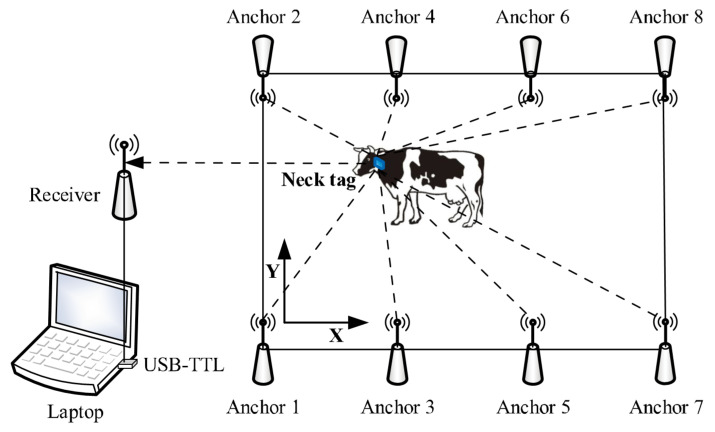
Working mechanism of the data acquisition system.

**Figure 3 animals-10-01160-f003:**
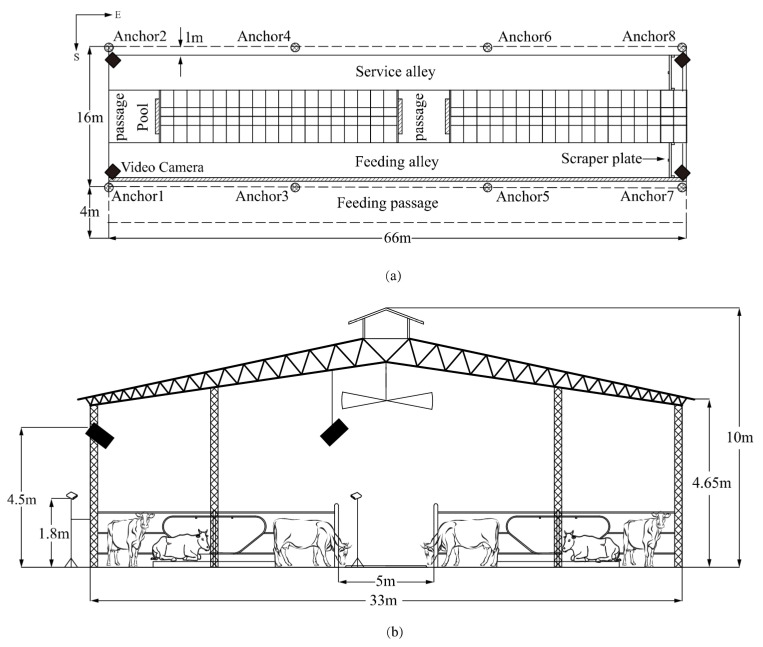
Plan and section of the studied area in the barn (**a**) Plan (**b**) Section.

**Figure 4 animals-10-01160-f004:**
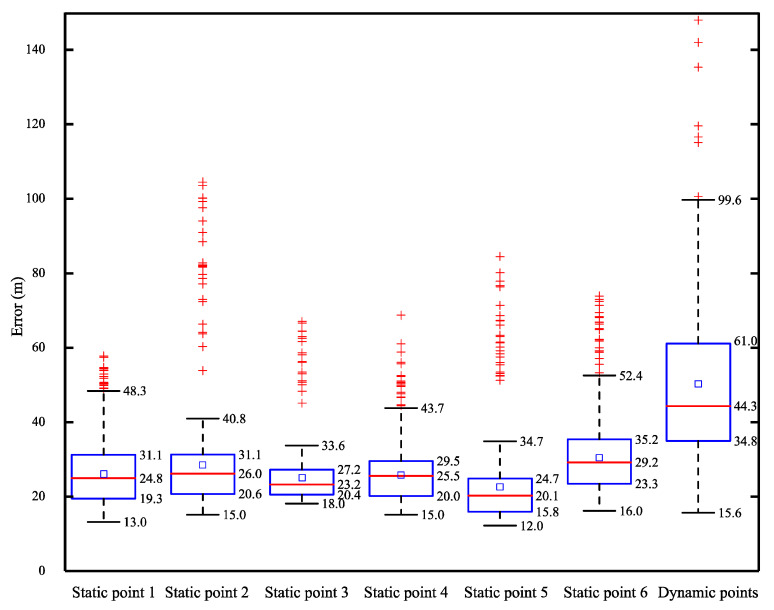
The boxplot of planimetric location errors of neck tag at static and dynamic points.

**Figure 5 animals-10-01160-f005:**
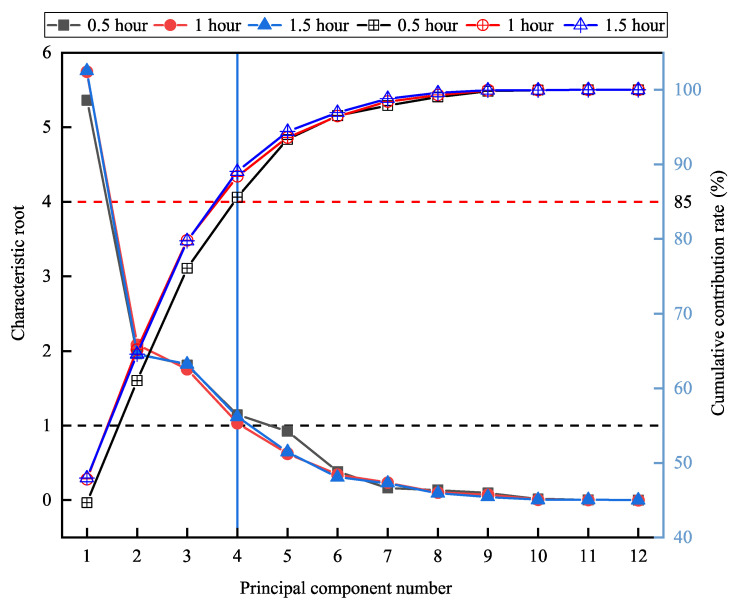
The characteristic roots and cumulative contributions of principal components under three time windows.

**Figure 6 animals-10-01160-f006:**
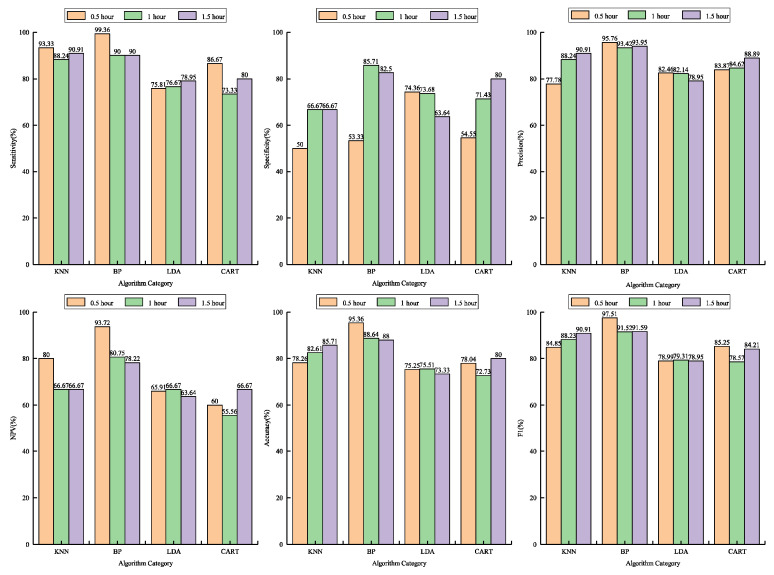
The performance of these four machine-learning algorithms for estrus detection.

**Table 1 animals-10-01160-t001:** Cows monitored and their estrus status and their stage of production.

Cow Number	Estrus Status	Parity	Days in Milk	Milk Yield/kg
1388	N	1	194	19.8
1391	N	1	168	20.6
0938	Y	Heifer	-	-
103	Y	3	62	23.1
105	Y	2	325	16.4
069	Y	2	65	40.4
123	N	1	539	18.6
2853	Y	2	287	26.3
1378	Y	2	261	12.1
118	Y	2	465	11.9
2960	N	3	52	28.5
3660	N	2	120	33.8

**Table 2 animals-10-01160-t002:** Scoring scale for observed estrus behavior in cows.

Estrous Signs	Points
Flehmen	3
Restlessness	5
Sniffing the vulva of another cow	10
Mounted but not standing	10
Resting with chin on the back of another cow	15
Mounting other cows (attempt)	35
Mounting head side of other cows (attempt)	45
Standing heat	100

**Table 3 animals-10-01160-t003:** Indexes used for assessing the performance of the four estrus detection algorithms.

Parameter	Calculation (%)	Definition
Sensitivity	TP/(TP + FN)	Proportion of identified estrus events among all estrus events
Specificity	TN/(TN + FP)	Proportion of identified non-estrus events among all non-estrus events
Precision	TP/(TP + FP)	Proportion of detected estrus events among all generated alerts
NPV	TN/(TN + FN)	Proportion of true non-estrus events among all detected non-events
Accuracy	TP + TN/(TP + TN + FP + FN)	Proportion of identified events among all events
F1 Score	(2 × TP)/(2 × TP + FP + FN)	Harmonic means of precision and sensitivity

**Table 4 animals-10-01160-t004:** Statistics of planimetric location errors for static and dynamic points.

Measured Points	Statistical Values (m) before Data Cleaning					Statistical Values (m) after Data Cleaning				
	Min ^a^	Max ^b^	Mean ^c^	Std ^d^	CV ^e^	Min	Max	Mean	Std	CV
Static point 1	0.13	0.58	0.26	0.09	0.35	0.13	0.48	0.25	0.07	0.28
Static point 2	0.15	1.04	0.28	0.14	0.50	0.15	0.41	0.25	0.06	0.24
Static point 3	0.18	0.67	0.25	0.08	0.32	0.18	0.34	0.24	0.04	0.17
Static point 4	0.15	0.69	0.26	0.08	0.31	0.15	0.44	0.25	0.06	0.24
Static point 5	0.12	0.84	0.22	0.11	0.50	0.12	0.35	0.20	0.05	0.25
Static point 6	0.16	0.74	0.30	0.10	0.33	0.16	0.53	0.29	0.07	0.24
Dynamic points	0.16	1.48	0.50	0.23	0.46	0.16	1.04	0.47	0.18	0.38

^a^ Minimum error; ^b^ Maximum error; ^c^ Mean location error; ^d^ Standard deviation; ^e^ Coefficients of variation.

**Table 5 animals-10-01160-t005:** Location performance of neck tag for static and dynamic points considering the metrics.

Measured Points	BODC ^a^			AODC ^b^			AEAODC ^c^		
	I ^d^	II ^e^	III ^f^	I	II	III	I	II	III
Static point 1	0	1	0.26 ± 0.09	0.04	0.96	0.25 ± 0.07	0.05	0.95	0.25 ± 0.07
Static point 2	0	1	0.28 ± 0.14	0.05	0.95	0.25 ± 0.06	0.08	0.92	0.25 ± 0.06
Static point 3	0	1	0.25 ± 0.08	0.04	0.96	0.24 ± 0.04	0.06	0.93	0.23 ± 0.04
Static point 4	0	1	0.26 ± 0.08	0.04	0.96	0.25 ± 0.06	0.05	0.95	0.24 ± 0.06
Static point 5	0	1	0.22 ± 0.11	0.05	0.95	0.20 ± 0.05	0.09	0.91	0.19 ± 0.05
Static point 6	0	1	0.30 ± 0.10	0.04	0.96	0.29 ± 0.07	0.06	0.94	0.28 ± 0.07
Dynamic points	0	1	0.50 ± 0.23	0.03	0.97	0.47 ± 0.18	0.08	0.92	0.45 ± 0.15

^a^ Before outlier data cleaning; ^b^ After outlier data cleaning; ^c^ Acceptable errors after outlier data cleaning; ^d^ Error rate; ^e^ Precision; ^f^ Mean ± Std.

**Table 6 animals-10-01160-t006:** Loads of the first four principal components for the three time-window lengths of 0.5, 1, and 1.5 h.

Behavioral Metrics	0.5 h				1 h				1.5 h			
	PCA1	PCA2	PCA3	PCA4	PCA1	PCA2	PCA3	PCA4	PCA1	PCA2	PCA3	PCA4
Duration of standing	0.348	0.003	0.012	−0.484	0.359	0.041	−0.042	−0.415	0.343	−0.075	0.067	−0.469
Duration of lying	−0.378	−0.112	0.075	0.362	−0.378	−0.102	0.102	0.307	−0.374	0.101	−0.117	0.344
Duration of walking	0.384	−0.210	−0.030	0.292	0.369	−0.238	0.037	0.233	0.373	−0.007	−0.229	0.240
Duration of feeding	0.079	0.571	−0.347	0.118	0.110	0.509	−0.367	0.074	0.090	−0.212	0.609	0.118
Duration of drinking	0.090	0.328	0.601	0.107	0.074	0.411	0.559	0.120	0.054	0.610	0.294	0.071
Switching times between activity and lying	−0.096	−0.031	0.268	−0.297	−0.161	−0.055	0.326	−0.576	−0.167	0.359	0.028	−0.421
Steps	0.377	−0.211	−0.032	0.311	0.364	−0.238	0.032	0.252	0.368	−0.014	−0.229	0.264
Displacement	0.382	−0.201	−0.025	0.305	0.368	−0.231	0.037	0.248	0.373	−0.005	−0.218	0.258
Average velocity	0.355	0.091	0.039	−0.414	0.367	0.099	0.053	−0.363	0.371	0.049	0.145	−0.328
Walking times	0.105	0.565	−0.320	0.159	0.133	0.496	−0.355	0.096	0.159	−0.239	0.550	0.263
Feeding times	0.351	−0.045	0.001	0.037	0.339	−0.094	0.092	−0.060	0.332	0.187	−0.138	−0.109
Drinking times	0.132	0.306	0.578	0.220	0.116	0.354	0.552	0.026	0.131	0.586	0.172	0.289

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
