# Peer review of "Machine-Learning Techniques Can Enhance Dairy Cow Estrus Detection Using Location and Acceleration Data"

_animals, 2020, doi:10.3390/ani10071160_

Round 1
Reviewer 1 Report
The problem statement was well described and the authors put the study into perspective nicely.
There were some minor formatting issues - eg Table 3 could be tidied up to prevent single letters being orphaned on to the next line.
This might have been an issue only with my pdf but the authors can check.
One of the things that did not come out clearly enough for me, and perhaps you could consider commenting, is the fact that the paper claims to have made a significant improvement over other reported methods for oestrus detection by using location. Much is made about the precision of the location measurement but its role in the model was not really all that clear. Was it used as a means of detecting walking etc and in which case, other approaches would be equally applicable? I remained a little unclear of the exact use of the location data. If the location was used to identify feeding or drinking then is it these behaviours that are important and the location was only a vehicle to access them.
In line 306 you comment ‘Moreover, it was remarkable that the upper and lower quartiles of neck tag at dynamic points were 2.04 times and 1.75 times of the average value at static points, respectively’. I was not sure why this was remarkable – it is possibly anticipated that an active node would be less precisely located.
ne of the issues with a study like this is that there were a limited number of animals and therefore the models have not been sufficiently tested for the conclusions to be as robust as claimed. If the same approach was adopted in a farm elsewhere would the results be the same. I am relaxed about this point because I understand the experimental challenges. Perhaps the authors can comment.
On the same point, given that there were only 12 cows, why did the authors choose the methodology they described to truth oestrus events. Progesterone sampling is widely considered as a gold standard and other assessment methods are open to interpretation.
Overall I was happy that the paper be published but perhaps the points above could be reflected on.
Author Response
Thanks for your suggestion. The responses are attached.

Reviewer 2 Report
Estrus detection is one of the most challenging topics in dairy production and a great opportunity for technological applications.
Introduction
L48-49: Please add a reference
L53-54: Estrus detection always requires a behavioral component, as such progesterone levels can’t be the “estrus detection gold standard”.
L54-55: In-line progesterone analysis (Herd Navigator) can monitor individual cows in a feasible manner considered reword the sentence.
L89: “of” instead of “for”
L86-94: Please clarify the term “location”, cow’s location in the barn or sensor location within cow’s movement?
L92-94: replace the term better to a more specific term such as “higher accuracy” or “higher estrus detection rate”
Experimental setup and data collection
L115: replace “a” for “the”
L126: to a mobile device
L140: “Fibre-board”
L143-145: Need to clarify if the 353 cows in the herd were open/cyclic other ways the possibility of choosing a pregnant cow were very high.
L147-148: Did you collected any physiological indication that the cows were in estrus? E.g. ovarian mapping, progesterone levels, estradiol concentrations etc. If so, you need to provide that information.
Table 1. By looking at the information provided in Table 1 (days in milk) you most likely selected pregnant cows. The most critical timing to detect estrus is between 60 to 100 days in milk to reduce de calving interval as you mentioned in the introduction section. Nine out of your 12 cows were beyond 100 days in milk. Can you provide the power statistics behind the 12 cows you selected (n=12).
L161: an individual
L162-163: Please provide inter-observer reliability level (e.g. Kappa coefficient).
Figure 2: Figure description appear blurry
Results
Figure 4. Can’t be read
Figure 6. Can’t be read
L222: “cross-validation”
L287: Please describe the outlier data-cleaning rule applied in this study in the data analysis section.
L329: “three-time”
L350-353: Please provide the specificity for each method.
Discussion
L369-378: The relationship between sensitivity and specificity is significant to understand the accuracy of estrus detection. For example, a sensitivity of 90% does not represent a high accuracy if the specificity is 10% (high number of false positives). Please rewrite the paragraph for clarity.
L383:”ultra-high”
L393: “showed”
L399: “multiple complex”
L399-403: Machine learning methods should improve automatically by experience (overtime). How is machine learning apply to the study? How many estrus occurrences per cow were observed in order to the algorithms learn from previous estrus data?
L420: “in” instead of “by”
Author Response
Thanks a lot for your suggestions. The responses are attached.

Round 2
Reviewer 2 Report
The authors have adapted the manuscript according to my remarks. Minor edits required.
Line 145-146: Not clear the sentence “the statistical power of days 146 in milks was 0.676”. The power statistics calculate the number of experimental units (n-value) needed in an experiment with a given alpha (i.e. 0.05).
I recommend providing a better resolution in Fig. 2, Fig. 4 and Fig. 6. Especially for Fig. 6 since it presents the accuracy of the algorithms tested.
Line 387: Avoid terms like good/bad
